# From Virtual Reality to Regenerative Virtual Therapy: Some Insights from a Systematic Review Exploring Inner Body Perception in Anorexia and Bulimia Nervosa

**DOI:** 10.3390/jcm11237134

**Published:** 2022-11-30

**Authors:** Clelia Malighetti, Maria Sansoni, Santino Gaudio, Marta Matamala-Gomez, Daniele Di Lernia, Silvia Serino, Giuseppe Riva

**Affiliations:** 1Department of Psychology, Università Cattolica del Sacro Cuore, Largo Gemelli 1, 20100 Milan, Italy; 2Department of Neuroscience, Functional Pharmacology, Uppsala University, Husargatan 3, 75237 Uppsala, Sweden; 3Department of Biomedicine and Prevention, University of Rome Tor Vergata, Viale Montpellier 1, 00133 Rome, Italy; 4Department of Psychology, Mind and Behavior Technological Center, University of Milano-Bicocca, Piazza dell’Ateneo Nuovo 1, 20126 Milan, Italy; 5Applied Technology for Neuro-Psychology Lab, IRCCS Istituto Auxologico Italiano, Via Magnasco 2, 20149 Milan, Italy; 6Humane Technology Lab, Università Cattolica del Sacro Cuore, Largo Gemelli 1, 20100 Milan, Italy

**Keywords:** regenerative medicine, inner body perception, anorexia nervosa, bulimia nervosa, proprioception, interoception, vestibular system

## Abstract

Despite advances in our understanding of the behavioral and molecular factors that underlie the onset and maintenance of Eating Disorders (EDs), it is still necessary to optimize treatment strategies and establish their efficacy. In this context, over the past 25 years, Virtual Reality (VR) has provided creative treatments for a variety of ED symptoms, including body dissatisfaction, craving, and negative emotions. Recently, different researchers suggested that EDs may reflect a broader impairment in multisensory body integration, and a particular VR technique—VR body swapping—has been used to repair it, but with limited clinical results. In this paper, we use the results of a systematic review employing PRISMA guidelines that explore inner body perception in EDs (21 studies included), with the ultimate goal to analyze the features of multisensory impairment associated with this clinical condition and provide possible solutions. Deficits in interoception, proprioception, and vestibular signals were observed across Anorexia and Bulimia Nervosa, suggesting that: (a) alteration of inner body perception might be a crucial feature of EDs, even if further research is needed and; (b) VR, to be effective with these patients, has to simulate/modify both the external and the internal body. Following this outcome, we introduce a new therapeutic approach—Regenerative Virtual Therapy—that integrates VR with different technologies and clinical strategies to regenerate a faulty bodily experience by stimulating the multisensory brain mechanisms and promoting self-regenerative processes within the brain itself.

## 1. Introduction

Despite advances in our understanding of the behavioral and molecular factors that underlie the onset and maintenance of eating disorders (EDs), it is still necessary to optimize treatment strategies and establish their efficacy. In this context, over the past 25 years, Virtual Reality (VR) has provided creative treatments for a variety of ED symptoms, including body dissatisfaction, craving, and negative emotions [1,2,3,4]. VR represents indeed an advanced imaginal system, able to generate emotions as if people were undergoing the same situations in real life [5,6]. Thanks to VR, clinicians may provide controlled exposure therapy (i.e., VR exposure) to their patients [7,8,9], offering a safe space in which to undergo experiences [10] or stimuli that are critical for the improvement of patients’ clinical conditions [8,9,11,12,13]. This characteristic enables patients to feel present in the virtual environment as if coping with the stimuli in real life [14], demonstrating to be particularly effective for exposure treatments [2]. In comparison to in vivo exposure (e.g., implemented through Cognitive Behavioral Therapy—CBT, guided imagery, etc.), VR offers a higher level of control and safety, permits the inclusion of contextual and proximal cues, prevents unforeseen events during exposure and helps to customize exposure to the needs of each patient, thereby lowering any treatment resistance and boosting motivation. Occasionally in vivo exposure can indeed be complex (e.g., it may be hard to uphold the required standards of safety and confidentiality when exposure is undertaken in a real-world setting, the time to travel to the exposure location may be long, there could be poor control over the stimuli, etc.) [10]. These restrictions can be somewhat circumvented by exposure in the clinic, although this method only permits exposure to proximate signals (e.g., meals), not to contextual cues (e.g., kitchen). Imagery is a second in vivo exposure option that is often carried out when facing EDs. However, if on the one hand imagery exposure addresses some of the aforementioned drawbacks, on the other hand, it also requires a significant amount of cognitive effort and may exhaust patients. As a result, there is a higher chance that patients will use avoidance tactics: clinicians, in fact, cannot fully control the scenario that patients are imagining [10]. When compared to imagery exposure, VR stimulates a variety of sensory modalities (e.g., auditory and visual), making it easier for participants who have trouble picturing scenes to participate. Additionally, since clinicians can see what the patient is seeing at any given time, VR aids in the identification of the stimuli that trigger a given emotional response [10]. VR-based cue exposure therapy (i.e., VR-CET) has proven greater effectiveness than CBT in decreasing binge and purge episodes in individuals with bulimia nervosa (BN) and binge-eating disorder (BED), showing a higher reduction in overeating episodes and a decrease in binge abstinence rates [15]. These findings are confirmed by other studies [16,17] which support the greater effectiveness of VR exposure for EDs when compared to in vivo one.

Recently, different researchers have suggested that EDs may reflect a broader impairment in multisensory body integration [18,19,20,21]. According to the Allocentric Lock Theory [22,23,24,25], patients suffering from Anorexia Nervosa (AN) are trapped in an outdated and negative memory of the body that cannot be changed even after a rigorous diet or significant weight loss: these patients are therefore prevented from updating their stored representation of the body (third-person perspective—offline) with new information coming from real-time perception-driven inputs (first-person perspective—online) [9,26,27,28,29]. Following this theory, a new VR technique called body swapping illusion [26,28,29] has been preliminary used as a clinical tool for EDs [29,30]. Using synchronous multisensory stimulation, body swapping induces the illusory experience of owning a virtual body: the perception of viewing an entire virtual body from a first-person perspective enables the participants to perceive the virtual body as their real one [31]. This methodology helps to reduce body-size overestimation in patients suffering from EDs, particularly AN [9,26,28,29]. However, differently from other clinical contexts (i.e., pain treatment) where the body swapping illusion is clinically effective [32], the existing results in EDs are disappointing: the effects of the VR experience are only temporary and tend to disappear in just a few hours after the treatment [33].

In this paper, we use the results of a systematic review that employs PRISMA guidelines and aims at exploring inner body perception in EDs, on the one hand, to understand the role played by deficits of inner body perception in the etiology of Eds; on the other, to use the results of this analysis to enhance the effects of VR-induced body modifications and propose a new approach to treat EDs.

### Inner Body Perception in Eating Disorders

Inner body perception is an umbrella term that encompasses primarily interoception, proprioception, and the vestibular system [34].

Interoception is “the sense of the physiological condition of the body” [35] and it is involved in a wide range of subjective experiences and fundamental aspects of bodily experience, such as body ownership [36] and self-awareness [37]. Particularly, Garfinkel et al. [38,39] distinguished and operationalized three different aspects of interoception: Interoceptive Accuracy (IAc), Interoceptive Sensitivity (IAs), and Interoceptive metacognitive Awareness (IAw). Specifically, IAc represents the ability to perceive inner bodily sensations, such as heartbeat; IAs represents the cognitive beliefs regarding the perception of the body, measured through self-report instruments; and IAw assesses the extent to which confidence predicts accuracy [38]. Proprioception is the sense of body position and movement [40,41], while the vestibular sense is intimately related to the inner experience of having a body [42], maintaining its orientation in the surrounding space thanks to the ability to provide continuous information about the body position [43,44].

There is evidence that multisensory integration may be disturbed in EDs e.g., [45,46], causing a mismatch between how the body is perceived and what the body is physically like [47]. Specifically, an impaired capacity to accurately sense, process, and integrate body signals has been observed in individuals with EDs [48], manifesting as a disturbance in bodily experience [34]. Poor ability to correctly perceive sensation from the inner body—a core element of multisensory impairments [34]—could be connected with observed deficits in coherently integrating input arising from within the body with the metacognitive perceptions of the body itself. Some studies have postulated that aberrant interoception—or the perception and integration of signals relating to body homeostasis (e.g., hunger, heartbeat, respiration)—might contribute to AN symptoms, including body image distortion, extreme restriction despite starvation and alexithymia [49,50,51,52]. In support of this notion, several lines of evidence have reported alterations in neural responses to taste stimuli, in individuals both with active symptomatology and in remission [53]. This deficit might be interpreted as dysfunctional integration of bodily information supporting a recent framework proposed by Riva and Dakanalis [20]. This model suggests that patients with AN are characterized by multisensory integration deficits that could affect the ability to properly relate the internal bodily signals with their positive or negative implications [20].

Along with this, numerous studies have now reported variations in the right parietal lobe function in EDs [54,55,56,57,58], suggesting altered proprioceptive perception related to body image representation. For example, Grunwald et al. [59] showed deficits in haptic perception and tactile-visual transformation in patients with AN, as well as diminished parietal activation during a task, suggesting proprioceptive integrative deficits in the parietal lobes. Similarly, Mohr et al. [60] conducted an fMRI study of body size estimation in AN patients and found evidence that body size overestimation may be related to issues with the retrieval of a multimodal body schema stored in the precuneus/posterior parietal cortex. This evidence suggests that parietal dysfunctions could be connected to body schema disturbances and that these kinds of alterations can also induce deficits in spatial orientation processes [61,62]. Several authors suggest the use of neurofeedback [63], invasive [64] and non-invasive brain stimulation techniques [64,65] to target the altered inner body perception of individuals with EDs. However, to date, no trials employing such methodologies have been implemented in this clinical population. The only technique that has been tested on patients with EDs is the cold-water caloric vestibular stimulation (CVS) [66]. CVS activates key nodes of the anterior cingulo-insular network (aCIN), altered in a wide variety of psychiatric and neurological conditions (e.g., EDs) [67], promoting vestibular neuromodulation. CVS works by performing a cold-water caloric vestibular stimulation of the ears. Specifically, external auditory canals are warmed or cooled using air or water irrigators. Temperature changes that are both warming and cooling cause the endolymphatic fluid in the semicircular canals to change in density, which in turn causes convection currents that cause cupular deflection, alter the tonic firing rate of the vestibular nerves and cause vestibulo-ocular reflex or horizontal nystagmus [68]. Schonherr and colleagues [66], revealed that after CVS (on the left and right ears), patients with AN reported a significantly smaller estimation of thigh width than before, closer to the real measurement. According to these authors, the Body-Perception-Index (BPI) reduced dramatically, too. Nevertheless, although these results are promising, they referred to only a few patients and no further trials have been implemented to replicate these findings on a bigger sample. For this reason, understanding how inner body perception is altered in EDs is an essential step to appropriately support patients, developing adequate interventions able to target not only their cognitive and emotional processes but also their bodily correlates. This systematic review represents, therefore, the first step to achieving this goal: before developing such interventions it is, indeed, essential to collect information to clearly define and fully understand if and how alterations in inner body perception are related. To reach this goal, the authors investigated the domains connected to inner body perception (i.e., interoception, proprioception, and vestibular systems) focusing on the tasks used for assessing such dimensions, as well as on the primary outcomes, in order to explore possible alterations in the two main ED clinical clusters: AN and BN.

## 2. Methods

A systematic review of scientific literature was performed to identify studies that reported assessment of inner body perception in individuals with AN and BN. To offer a broad panoramic of the current state of the art on the topic, we did not define a beginning year of publication for the articles to be included. A review protocol following the Preferred Reporting Items for Systematic Reviews and Meta-Analysis (PRISMA) guidelines [69] was compiled.

### 2.1. Data Sources and Search Strategy

Data sources of relevant publications on experimental studies were collected on the 5th of August 2022 through a computer-based search in three high-profile databases: PubMed, Web of Science (Web of Knowledge), and PsycINFO. Each database was searched independently according to three specific iteration research strings: (Eating Disorder) OR (Anorexia) OR (Bulimia) AND (“Internal Body” OR “Body Sensation” OR “Proprioception” OR “Interoception” OR ”Interoceptive” OR “Vestibular” OR “Autonomic system” OR “Visceral” OR “Internal Perception” OR “Body experience”). To make this study repeatable in the future, detailed results of the search strategy are available in the Appendix A. The selection of these strings was made in an attempt to capture a broad range of features regarding bodily perception and EDs. Citations were retrieved independently for each iterative search crossing all databases. The complete list was exported and aggregated to remove duplicates and then imported into Rayyan [70] for the title and abstract screening. The list of studies selected for inclusion was also sent to leading experts in the field for suggestions and identification of any missing studies. As a result, one study [71] was screened and included in the review.

### 2.2. Study Selection and Inclusion Criteria

Inner body perception is considered to be composed of different sensory inputs: proprioceptive, interoceptive, and vestibular e.g., [34,72]. To be included in the review, studies were required to:(a)Investigate a sample of individuals that meet a current diagnosis of AN and BN, according to the Diagnostic and Statistical Manual of mental disorders (DSM) or International Classification of Diseases (ICD). In other words, studies in which the participants self-reported the diagnosis, used self-reported measures to identify participants’ diagnosis, or in which the diagnosis was not provided by a professional (e.g., a clinical psychologist) were not included in the systematic review. Both adults and adolescents with a current diagnosis of AN or BN were considered eligible. Studies that considered the participants’ sample as aggregated (e.g., reporting under the same category of EDs multiple diagnoses) were also not included in the systematic review;(b)Include a healthy control group (HC) or a population of normative values to compare the clinical group with;(c)Use tasks or instruments to evaluate interoception according to Craig’s definition [35,38]; or proprioception according to Blanke’s definition [40]; or vestibular perception according to Lopez’s definition [73]. Studies that employed self-reported questionnaires to assess such dimensions were excluded;(d)Use tasks that directly evaluate one or more sensory domains. Interoceptive input was considered present when the task tested sensitivity to visceral activity [35]. Proprioceptive input was considered present when the task was based on a sensory judgment about limb and body position [40]. Vestibular input was considered present when the task tested the sensation of any change in balance, position, direction, or movement of the eyes, head, or body [73].(e)Use behavioral and cognitive tasks. Studies involving manipulation of the variable of interest (e.g., through medications or psychological interventions) were not included in the systematic review;(f)Be original articles: reviews, meeting abstracts, conference proceedings, notes, letters to the editor, research protocols, patents, editorials, books or chapters, and other editorial materials were not considered eligible for this systematic review;(g)Be quantitative studies: qualitative studies were not included;(h)Be in English, enroll humans (i.e., studies that use animals were excluded) and have an available full text.

A flow chart of the search strategy according to the PRISMA Flow Diagram is available in Figure 1.

### 2.3. Study Inclusion

Two reviewers (C.M. and M.S.) independently screened all non-duplicate titles and abstracts, searching for eligible articles. The same reviewers retrieved and analyzed the full text for all relevant articles, resolving discrepancies in opinions by consensus. D.D.L. was designated as the third reviewer to arbitrate potential differences in agreement.

### 2.4. Data Extraction

Two reviewers (C.M. and M.S.) independently extracted the following data: group sample, composition and gender; specific diagnosis; interoceptive, proprioceptive and vestibular task or instrument used for the assessment; construct measured; and primary outcomes. Data are available in Table 1.

## 3. Results

Of 19,672 studies retrieved from PubMed, PsycINFO, and Web of Science, 6807 were non-duplicates. After screening all non-duplicate titles and abstracts, 6495 did not fit the preliminary inclusion criteria. Subsequently, the full text of 312 articles was retrieved and the studies were analyzed for the specific inclusion criteria. Of these 312 studies, 292 were excluded. Reasons for exclusion were lack of appropriateness of the study sample (e.g., no AN or BN diagnosis according to DSM or ICD, employment of an aggregated sample of EDs, etc.), no pertinence of the construct analyzed in the study (i.e., no proprioception, interoception or vestibular system examined), or no appropriate task involved (e.g., manipulation of the construct through medications, assessment implemented using self-reported questionnaires instead of tasks, etc.). Other reasons for exclusion were the absence of a control group, or of an available full text. Please see Figure 1 for more details about the inclusion/exclusion process. Therefore, only 20 articles met the inclusion criteria and were identified as suitable for our review. One additional paper [71] was also included upon suggestion by leading experts in the field, leading to a total of 21 papers included in the review.

In the following paragraphs, study characteristics and results will be presented. The Section 3 will focus on the assessment of proprioceptive, interoceptive, and vestibular perception, as well as on the primary outcomes of the studies. Detailed information about study characteristics, including sample, gender, diagnosis, task or instrument employed, assessed construct, and primary outcomes measured are presented in Table 1.

### 3.1. Study Characteristics

Table 1 shows study characteristics according to extraction parameters. Sixteen studies explored interoception in AN or BN [71,74,75,76,77,78,80,81,82,83,86,87,89,90,91,92], five investigated proprioception [71,79,84,88,93] and one vestibular signals [85]. Among these studies, one [71] assessed interoception and proprioception within the same work.

In terms of the sample, most studies compared patients with AN and an HC group [71,74,76,78,79,80,84,86,87,88,89,90,91,93]. One study [83] was a single case study on a patient with AN who was compared with a group of four HCs. Two studies [75,85] compared patients with AN, BN, and a group of HCs, one study [92] compared patients with BN with HCs and other two studies compared the clinical sample (i.e., AN) both with HCs and with recovered patients [77,81]. Lastly, one study compared patients with AN, HCs, and patients reporting functional motor symptoms [82]. In general, the clinical samples of included studies ranged from one participant [83] to a maximum of 37 individuals [86,90], and all studies included female participants.

### 3.2. Tasks Employed to Assess Interoception, Proprioception, and Vestibular Processes

#### 3.2.1. Interoception

Eight studies [74,82,83,86,87,89,90,91] investigated cardiac IAc [94] by using the heartbeat perception task [95]. This technique consists of silently counting heartbeats in a defined time frame (25 s, 35 s, 45 s, 100 s) without any external heartbeat information and focusing only on inner body perception. The heartbeats reported by patients are compared to the real heartbeats measured through an ECG or other appropriate medical equipment (e.g., pulse oximeter). Wollast et al. [91] repeated the task twice: while at rest and after listening to a song. This second task was used for emotional induction and was modeled after that used by Mayer et al. [96], who employed a sad piece of music to induce negative emotions, thus simulating, in this way, physiological reactions and modifications in the heart rate. Three studies [83,86,87] added measures for a comprehensive interoception assessment. Indeed, besides IAc, these authors also assessed IAw, asking the participant to provide a degree of confidence regarding their performance on the heartbeat perception task. To assess this, confidence ratings were used. In particular, following the heartbeat perception task, participants were asked to rate the level of confidence in their performance on a scale from 1 (least confident) to 100 (most confident), mostly using a Visual Analog Scale. Di Lernia et al. [83] also assessed the interoceptive buffer saturation index (IBs). This task, based on a verbal estimation of interoceptive tactile stimuli led with a specific device [83], aims to reversely evaluate the amount of interoceptive processing through distortions in the time perception of the stimuli [97].

Three studies [77,80,81] assessed interoception using Affective Touch [98,99,100,101]. In Bellard et al. [77] the Affective Touch consisted of Self and Other-directed Affective Touch video clips. Specifically, touch was delivered across five different body regions: non-C Tactile (CT)-innervated body site (i.e., palm) vs. CT-innervated body sites (i.e., ventral forearm, upper arm, cheek, and back) with three different speeds: static (0 cm/s), slow (5 cm/s) and fast (30 cm/s). After watching each video, participants answered two questions using a 100-point VAS scale (i.e., 0 = very unpleasant, 100 = extremely pleasant): “How much would you like to be touched like that?” (Self-directed touch) and “How pleasant do you think that action was for the person being touched?” (3rd person perspective—Other-directed touch). The task was implemented among patients with AN, recovered patients, and HCs to analyze differences in interoception. In their study, Crucianelli et al. [80] measured the perceived pleasantness of stroking touches applied to the forearm of patients with AN and HCs while participants were looking at photos of young women’s faces displaying smiling, rejecting, or neutral expressions. Tactile simulations were conducted with CT afferents-optimal (3 cm/s) and non-optimal (18 cm/s) velocities while simultaneously displaying the photos. Participants were guided to leave the stimulated arm inside a box open on two opposite sides: this detail guaranteed adequate delivery of the touch while at the same time preventing the participant from receiving visual feedback of the tactile stimuli. Lastly, in their subsequent work, Crucianelli et al. [81] added to the two stroking touches used in their previous study [80] (3 and 18 cm/s, respectively, CT afferents, optimal and non-optimal) and other tactile stimulations administered at five different speeds: an additional CT-optimal stroking touch (6 cm/s), one borderline touch (9 cm/s) and another not CT-optimal stroking touch (27 cm/s). Due to the high number of stroking touches implemented, stimulation was cycled between the two locations on the participant’s forearm to prevent habituation and fatigue of CT fibers. To avoid visual feedback, in this study participants were blindfolded throughout the task. As with Bellard et al. [77], the study by Crucianelli et al. [81] compared persons with AN, recovered patients, and HCs.

Among the studies that focused on interoception, two [75,78] explored gastric interoception [94]. Aschenbrenner et al. [75] utilized the “sniffin’ sticks” test battery [102] and the “taste strip” test kit [103] for the assessment of patients with AN, BN, and HCs’ olfactory and gustatory functions, respectively. The sniffin’ sticks test consists of a felt-tip pen-like odor dispenser that is used for an ortho-nasal examination of olfactory function and comprises three tests of olfactory functions: tests for odor threshold, odor discrimination, and odor identification. The taste strip test [103] consists of the administration of spoon-shaped filter paper strips impregnated with four taste qualities presented with increasing concentrations and placed on the left and right side of the anterior third of the tongue, resulting in a total of 32 trials. Participants in Aschenbrenner’s study [75] had to identify the taste from a list of four descriptors. For each correct answer, patients received one point, which accounted for a maximum score of 32. The study by Brown et al. [78] employed the Water Load Task (WLT) to noninvasively assess gastric interoception in patients with AN and HCs. Participants sat in a half-supine position and were asked to drink water until their stomach was “completely full” (i.e., entirely filled with water). In total, 1.5 L of water was available for drinking. The exercise stopped when participants raised their hands to communicate that they had reached complete fullness. The task was halted by personnel after five minutes if the participant had not raised their hand. Researchers registered how long it took participants to reach complete fullness and, at the conclusion of the exercise, participants were asked to estimate how much water they had drunk. Precisely, participants were instructed to use a 1.5 L carafe completely full of water to pour into another empty one the quantity of water they believed they had drunk. Positive numbers indicated an overestimation of water drank since the accuracy of this estimate was computed by deducting the water estimated from the water actually consumed. Individuals were then asked to rate their level of confidence in this estimate using a range of 0 to 100.

Finally, three studies [71,76,92] investigated acute pain, which is a primary interoceptive perception [94]. Bär et al. [76] provided an assessment that included heat pain measures collected from individuals with AN and HCs. The heat pain thresholds were assessed on both arms by an ascending method of limits with a contact thermode attached to the left or right volar wrist. To determine heat pain thresholds, individuals were asked to press the stop button immediately when thermal perception had become painful. Goldzak-Kunik et al. [71] investigated cold pain in patients with AN compared to a control group of healthy people using an ice cube as a pain temperature stimulus. Three trials of 15, 30, and 45 s in duration respectively were administered with 15 min of breaks to separate the trials. Once each trial was concluded, participants rated cold, unpleasantness, and pain using three different VAS scales. Yamamotova et al. [92] assessed the thermal pain threshold latency among patients with BN and HCs using the Analgesia Meter (IITC Life Science USA Model 33), which employs radiant heat of constant intensity to an area of 1 cm^2^. The assessment was carried out under six consecutive conditions, three at rest and three under stress: rest I, mental arithmetic task, rest II, eating sweet food, rest III, and cold pressor test.

#### 3.2.2. Proprioception

In our review, we found that five studies investigated proprioceptive perception in individuals with AN and HCs [71,79,84,88,93]. Specifically, Epstein et al. [84] analyzed the proprioceptive aspects of body perception through the “proprioception test” and the “right-left orientation test”. The proprioception test assesses the capacity to locate one’s body parts in space. The task consists of asking the subject to touch, on verbal command, specific points on the body, without any visual input. In particular, participants in Epstein’s study [84] were instructed to use their right index finger to touch ten different points on their bodies. Importantly, participants did not have to move any part of their body other than their right arm and hand. Regarding the second task used by Epstein et al. [84], the right-left orientation test evaluates three features of right-left orientation (i.e., orientation toward one’s own body, toward a confronting person, and the combined orientation of the previous two).

Zopf et al. [93] used the Rubber Hand Illusion (RHI) paradigm instead to explore if the body location perception of patients was influenced differently by two types of multisensory conflicts: visual-proprioceptive hand location and visual-tactile touch synchrony. This paradigm involves, in fact, the interaction between touch, vision, and proprioceptive perception of the body in space [104,105]. The RHI paradigm [106] consists of a perceptual illusion of feeling ownership of a fake hand and provides a quantitative measure of embodiment. During the RHI paradigm, participants feel as if a fake hand belongs to them due to synchronous visuo-tactile stimulation of both a fake rubber hand, located within the visual field of the participant, and the participant’s real hand, located outside the visual field of the participant. This task provides two outcome measures: proprioceptive drift [107] and the level of ownership illusion [106]. Proprioceptive drift is calculated by asking participants to indicate the position of the tip of their left index finger prior to and following each visuo-tactile stimulation (performed in asynchronous and synchronous conditions). The difference between hand estimates before and after inducing the RHI is the “proprioceptive drift” [107]. The level of ownership illusion over the rubber hand is obtained using self-report questionnaires that provide a subjective measure of the illusion e.g., [106]. Zopf and colleagues [93] measured the effect of the illusion through the reaching responses toward visual targets and the movement endpoints and the extent of the illusion using explicit bodily judgments with a set of evaluations adapted from existing RHI questionnaires [106,108]. Case et al. [79] utilized a size-weight illusion battery to evaluate visual and proprioceptive information, instead. The illusion used in the task consists in leading the subject to underestimate the weight of a larger object when compared to a smaller object of identical shape and weight. Mergen et al. [88] used the One-Point-Localization task to assess the distorted representation in AN through two experiments. This task is an adaptation of the localization paradigm, which consists of asking participants to localize a tactile stimulus placed on their skin on a screen that showed a live image of the touched body part. The aim of experiment 2 was to extend the results from experiment 1 by exploring differences in the task on neutral and sensitive body parts [88]. During both experiments, the investigator touched the participants’ back or abdomen with a rubber stick, and participants were asked to click the mouse as soon as they perceived the body stimulation. The click generated a photograph of the back/abdomen which was subsequently presented to participants with the outlines obscured. At this point, participants were asked to indicate the position of the perceived touch on the image and confirm this position with a second mouse click [88]. Lastly, Goldzak-Kunik et al. [71] used a kinesthesia task to examine sensory dimensions relevant to spatial and motion aspects of body size perception. A vertical handle was put in the sloping rails of an apparatus at around chest height and participants were instructed to estimate the relative height of each hand, holding it while wearing blindfolds. As the slope changed in a set pattern of rises and declines, the left hand was lifted or lowered.

#### 3.2.3. Vestibular System

Only one study examined vestibular signals [85]. Precisely, Fontana et al. [85] investigated the postural stability of individuals with AN, BN, and HCs, through the analysis and quantification of their postural strategies under standardized quiet-standing conditions: with eyes open (EO) and closed (EC). The acquisition duration was 60 s and participants were asked to present the feet spread apart at shoulder width. The kinematics (or segmental) method was adopted to quantify the Center of Mass (CoM) position, which in the study by Fontana et al. [85] was conceptual, with no direct measure to locate it in space. A passive marker optoelectronic system (Vicon 460) was employed and the CoM position was estimated using the positions of three-dimensional markers and a biomechanical model.

### 3.3. Primary Outcomes in Anorexic and Bulimic Patients

The included studies showed different results that will be carefully evaluated in the discussion of this review.

#### 3.3.1. Interoception Outcomes

Eight studies [74,82,83,86,87,89,90,91] used the heartbeat perception task [95] for the assessment of cardiac interoception in patients with AN. Di Lernia et al. [83], Pollatos et al. [89] and Wollast et al. [91] showed deficits in IAc in patients with AN compared to controls. Specifically, Di Lernia et al. [83] performed a complete interoceptive assessment before and after an outpatient rehabilitative hospital program and the results showed severe deficits in accuracy, buffer saturation, and sensitivity in the patient compared to the control group. Pollatos et al. [89] displayed that compared to HCs, people with AN exhibited lower IAc during self-focus. In line with these outcomes, also Wollast et al. [91] found a deficit in IAw in the patients suffering from AN compared to the HCs, both at rest and when an emotional context was induced. Ambrosecchia et al. [74], Demartini et al. [82], Kinnaird et al. [86], Lutz et al. [87] and Richard et al. [90] showed no differences between patients with AN and HCs in IAw, instead.

With regard to interoceptive touch sensitivity, all studies [77,80,81] revealed deficits in interoception borne by patients with AN. Crucianelli et al. [80] showed that individuals with AN perceived affective touch as less pleasant compared to HCs, suggesting that this reduced pleasantness may be at least in part related to a dysfunctional CT afferent system. In agreement with this result, also the study by Crucianelli et al. [81] indicated that both patients with AN and recovered participants anticipated tactile experiences and rated delivered tactile stimuli as less pleasant than HCs. However, this difference was not related to the CT optimality of the stimulation. Instead, variations in top-down beliefs, alexithymia, and interoceptive sensitivity predicted changes in how CT-optimal touch was perceived. As a result, tactile anhedonia in AN may last even after a generally successful recovery and it is associated with a taught, flawed top-down expectation of tactile pleasantness rather than a bottom-up interoceptive deficiency in the CT system. The study by Bellard et al. [77] also evidenced this when evaluating touch for self: women with AN and recovered patients compared to HCs rated CT-optimal touch as less pleasant than HCs, even if they did not differ in pleasantness ratings when evaluating affective touch for another person.

The studies that explored gastric interoception reported similar results. Specifically, Aschenbrenner et al. [75] showed that individuals with BN and AN exhibit lowered olfactory and gustatory sensitivities compared to HCs. These deficits improved with increasing BMI and decreasing eating pathology in the course of treatment. Moreover, Brown et al. [78] displayed that participants with AN tend to overestimate the amount of water consumed and report greater levels of pre and post-Water Load Task fullness compared to HCs. However, regarding this latter result, no group-by-time interaction was found, suggesting that overall, there were no significant differences in change of fullness between groups. Furthermore, individuals with AN also reported greater increases in negative affects pre to post Water Load Task compared to HCs, but confidence regarding consumption estimation was not different between the two groups.

Ultimately, results concerning pain perception are mixed. Bär et al. [76] showed that heat pain thresholds significantly increased in the acute state of AN and decreased after weight had been regained for six months. Yamamotova et al. [92] showed that thermal pain threshold latency is longer in patients with BN than in HCs and that the BN group has a significantly higher pain threshold under all six experimental conditions. Finally, Goldzak-Kunik et al. [71] did not show differences between the AN and the HC group in cold pain responses.

#### 3.3.2. Proprioception Outcomes

Proprioceptive perception has been investigated in patients with AN by five studies [71,79,84,88,93]: three of them point out a difference between proprioception in individuals with AN and healthy women, while the other two indicated no difference between groups. Case et al. [79] found that individuals with AN show a reduced size-weight illusion compared to controls, indicating a decreased capacity to combine visual and proprioceptive information. This alteration could lead to distorted body perception. Epstein et al. [84] demonstrated that patients with AN compared to an HC group reported significantly lower scores in the “right-left orientation test” at pre-treatment assessment and no significant differences at post-treatment. Zopf et al. [93] found a reduced influence of proprioceptive signals on hand location estimates in AN compared to controls. Contrary to these results, Mergen et al. [88] revealed that patients with AN and HCs did not differ in the ability to accurately localize the tactile stimulus onto a visual presentation of the body. Furthermore, no differences were found between the performance at the back and the abdomen. However, both groups showed distorted perceptions in both experiments and for at least one body part. Goldzak-Kunik [71] also found no differences between individuals with AN and controls in the performance obtained in the “kinesthesia” task.

#### 3.3.3. Vestibular Outcomes

Fontana et al. [85] was the only study that assessed vestibular deficits in individuals with an ED. The results proved that women with BN are more unstable than HC individuals, showing significant differences in CoM anteroposterior excursions and length of the road, while patients with AN showed no significant differences from HCs.

## 4. Discussion

In the manuscript we suggest that it is possible to use technology (i.e., VR) to create simulative bodily experiences and that these experiences can alter the functioning of the body, triggering regenerative processes able to address complex pathologies. The aim of this systematic review was to explore studies that investigated whether inner body perception was altered in AN and BN in order to facilitate the development of clinical interventions targeting such dimensions through technology. Overall, the analyzed studies in this review show that inner body perception seems to be indeed altered in EDs, with different alteration patterns in AN and BN.

### 4.1. Interoceptive Deficits in Anorexia and Bulimia Nervosa

Results from the systematic review suggest that patients with AN might exhibit interoceptive perception deficits, as reported by lower accuracy scores on the heartbeat counting task compared to HCs. Specifically, individuals with AN showed difficulties in distinguishing actual interoceptive sensations from anticipated ones, particularly at low levels of bodily arousal, compared with HCs. A similar pattern was found in individuals with remitted AN: recovered patients reported altered neural activation during anticipation and receipt of sucrose tastes [109]. Some studies reveal indeed that habituation to fullness is protracted after eating [52], indicating that the return to homeostasis after state changes may also be impaired in people with AN. This alteration might reflect a dysfunctional integration of bodily information since lower IAc is associated with a higher malleability of body representations [36,110]. Along these lines, Berner et al. [109] suggest that a brain-based difficulty predicting and adapting to internal state shifts may contribute to the severity and persistence of AN. Support for this perspective can also be found in theories linking interoceptive prediction error to anxiety [111], associations between perceived sensory sensitivity and emotion dysregulation in AN [112], and the observed relationships among markers of AN severity and prefrontal and striatal hyperactivation after aversive interoception [109]. However, certain studies of this review did not find evidence for altered interoceptive heartbeat perception. Thus, moderating factors might contribute to such heterogeneity. One explanation of this inconsistency may be the variability of the samples included in these studies (inpatients, outpatients, smaller sample size, duration of the pathology, comorbidities, etc.), which results in heterogeneity in weight and treatment progress. Furthermore, individuals were evaluated at different stages of treatment, suggesting that treatment progress may have an impact when investigating interoceptive deficits in AN. In particular, interoceptive deficits might interact with weight gain or recovery periods, which can therefore be confounding factors and should be controlled for in interoceptive experimental studies. Specifically, Richard et al. [90] found that individuals who gained more weight and spent more time hospitalized showed higher IAs. This result indicates that interoceptive processes may be influenced by state-dependent factors and heterogeneity in treatment progress. However, there is evidence that interoception remain reduced in patients with AN at the end of treatment [89], suggesting that this alteration of bodily signals might be an ongoing risk factor for the maintenance of AN. Although heartbeat counting tasks are commonly used to assess interoception, it should be noted that there are methodological limitations to this approach [113]. For example, knowledge of one’s resting heart rate influences the accuracy of heartbeat counting tasks [114]. In addition, only around a third of participants can accurately count their own heartbeat at rest, which opens up the possibility that floor effects may explain some null findings [94,115]. Furthermore, in some cases the task might be perceived as difficult, leading patients with AN to a higher level of stress and arousal that affects performance.

The systematic review also identified experimental studies focused on assessing pain in individuals with AN. Because pain represents the first and primary interoceptive input, this specific afferent information maintains its value in disclosing the way the interoceptive system works in AN. In relation to pain processing in individuals with AN, our results showed an increase in pain thresholds in the acute phase of the disease that decreased six months after regaining weight. However, other studies found no differences in cold pain perception among people with AN. Our findings are in line with previous studies suggesting that individuals with AN, BN, or BED have elevated thresholds to thermally [116] and mechanically induced pain, but they do not exhibit similar alterations in their sensitivity to cold [117]. Several psychological and biological mechanisms have been associated with decreased sensitivity to pain in EDs, including impairments in emotional and cognitive processing such as alexithymia and dissociation, nutritional restrictions, decreased skin temperature, blood pressure, and broader dysregulation of the vegetative nervous system [118,119]. Furthermore, the reduced pain sensitivity found in AN might be associated with insular dysfunction [120]. Strigo et al. [121] showed that patients with AN have a reduced capacity to accurately perceive bodily signals [51,112], which seems to persist even after recovery. The observed mismatch between subjective experiences (ratings) and objective responses (brain activation) in AN suggests, therefore, abnormal integration processes and, possibly, a dissociation between reported and actual interoceptive states. Deficits in interoceptive perception might play an important role in the etiology and maintenance of EDs. This decreased pain sensitivity in AN might be due to a reduction in the ability to correctly perceive the inner body dimension. In relation to pain processing, the variability of the results might be explained by a lack of consistency of measures across studies (e.g., the use of heat vs. cold stimuli). Another reason could be a limitation in the methodology used for induction of thermal pain, as well as the lack of pain threshold and tolerance measures. Many studies have repeatedly found increased pain thresholds in individuals with EDs [116,118,122], whereas others have shown no differences in pain thresholds compared to HCs e.g., [123].

One of the modalities in which impairments were consistently associated with EDs was sensitivity in gastric interoception. Individuals with AN exhibit lowered olfactory and gustatory sensitivities. These deficits might be transferred to the perception of bodily signals in general, including the accuracy of bodily signals such as hunger and satiety. Our review, in fact, also showed that participants with AN drank significantly less water than HCs and reported greater increases in negative affects after the task (i.e., Water Load Task). The perception of fullness was greater in AN compared to HCs, but since there was no group-by-time interaction, overall there were no significant differences in change in fullness between groups. At present, it is not known whether these perceptual distortions are a determinant or a result of AN or whether they improve following successful treatment. Furthermore, this lack of satiety aversion is thought to be related to people who overestimate their visual self-image. Garner and Garfinkel [124] reviewed several studies that display how individuals with AN are less accurate in judging interoceptive sensations than HCs. Several lines of experimental inquiry have suggested indeed that patients with AN may misperceive internal experiences, particularly those related to satiety. Further, the more individuals with AN overestimate their body size, the less sucrose aversion they manifest. An analysis of sensations after eating indicates that patients with AN feel fuller before eating than HCs. Individuals with AN also reporte more postprandial bloating, nausea, and thoughts of food. These findings suggest that patients with AN may experience sensations associated with eating differently than HCs do. It is, therefore, possible that people with AN have an altered ability to recognize certain visceral sensations related to hunger, satiety, and pain, suggesting a reduced capacity to accurately perceive inner body signals. In this regard, the reduced pleasantness of tactile stimuli we found in our review among individuals with AN might be read in light of this reduction in the ability to integrate and accurately perceive inner body signals. These findings, however, might also be explained by the possibility that the observed decrease in pleasantness perception is a result of an effort to regulate (i.e., lessen) anxiogenic stimuli. Anxiety is quite common among people with EDs and is accompanied by enhanced activation of cognitive control in an effort to balance out the diminished limbic function (i.e., more strategic choices can compensate for the impaired ability to perceive interoceptive information). Therefore, the decreased physiological pleasure that we noticed could be an effort to cognitively regulate an “unwanted” stimulating experience (i.e., pleasant interpersonal touch) [125].

Regarding BN, our review showes that patients with BN present lower sensitivity to pain, which seems to be a stable phenomenon and persists under various experimental conditions. This result is also consistent with the finding that pain sensitivity remains low in women who are long-term recovered from BN [126]. Supporting this idea, a recent study by Pollatos and Georgiou [127] observed an abnormal integration of different interoceptive signals in patients with BN. It is, therefore, possible to assume that individuals with BN and AN have a reduced ability to correctly elaborate the probabilistic process connecting the different inputs from exteroceptive, proprioceptive, interoceptive, and vestibular sensory systems that are essential for body self-consciousness.

### 4.2. Proprioception in Anorexia and Bulimia Nervosa

The studies included in this systematic review found several impairments in AN compared to controls with regard to the proprioceptive component of inner body perception. These findings are in line with other researchessuggesting impairments in spatial orientation in AN [61,62,128]. There is evidence that individuals with AN show impaired spatial cognition and that those deficits might be related to poor awareness of interoceptive inputs [129]. Furthermore, AN is characterized by alterations in posterior parietal areas [130] that are also related to the egocentric spatial reference frame directly involved in spatial cognition [131]. In this view, the low capacity of individuals with AN to integrate egocentric and allocentric spatial reference frames related to alterations in posterior parietal areas [34] may explain impaired proprioceptive processing. Moreover, it is known that parietal cortex activity is linked to the processing of proprioceptive sensory information and the integration of multisensory body information to update body size and location information [132,133,134]. Furthermore, Zopf et al. [93] reported a decrease in proprioceptive signals on hand location in patients with AN compared to HCs, suggesting that individuals with AN are more influenced by external visual information and relatively less by proprioceptive information. This tendency in individuals with AN toward external visual body information could be attributed to differences in the processing of proprioceptive signals. In a haptic task without vision, in which active exploration of objects depends on proprioceptive body position, Grunwald et al. [59] showed impairments in the processing and storage of proprioceptive information in individuals with AN compared to HCs. Typically, where visual information is available, proprioceptive and visual hand location information is integrated to form hand location estimates [135]. However, in individuals with AN, there is a deficit in proprioceptive-visual integration that could result in vision becoming a more dominant source of information. In addition, the study conducted by Zopf et al. [93] revealed that in individuals with AN multisensory body perception changed: the proprioceptive signals decreased and the relative influence of external visual information increased for the perception of a body location. The authors [93] suggested that this tendency of patients with AN toward external visual body information is due to changes in proprioceptive signal processing. The recurrent changes in the physical body could potentially cause modifications in multisensory body perception in AN. In relation to localization, one study showed no differences between AN and HCs in the ability to localize the stimulus or between performances at different body parts [88]. However, in that study, both groups showed systematically distorted perceptions across experiments and for at least one body part. The authors suggested that focusing on localization instead of on body size or distance estimation could minimize the cognitive-affective influences [88]. Furthermore, even if patients with AN reported a significantly worse cognitive-affective body image compared to HCs, this did not affect the One-Point-Localization Task performance. These findings could suggest that body distortion in AN may be related also to the cognitive-affect component besides perceptual alterations [88]. In contrast with Zopf’s assumption of visual dominance over body location perception in AN [93], Case et al. [79] displayed that patients with AN have a cross-modal sensory integration deficit with a greater reliance on proprioceptive information, compared to HCs. The study suggests less influence of visual object information on the perception of heaviness in AN. One explanation of this impoverished visual process in AN might be malnutrition, which affects vision or sensory integration as has been shown by Mohr [136], or a preference for proprioceptive information. Altered proprioceptive information about the body could explain the over-evaluation of weight and size in AN and the distorted perception of body image [79]. Furthermore, as suggested by Case and colleagues [79], distortions are likelier to occur in proprioception than in vision since we have a proprioceptive sense primarily of our own bodies and those of others. Since AN primarily affects the sense of one’s own body, proprioception would seem a more likely candidate for a sensory disturbance. Hence, the altered multisensory integration could be explained by a different internal model of heaviness in individuals with AN that could generate different expectations based on visual information [79]. Reduced size-weight illusion (SWI) in individuals with AN fits with the emerging picture of interoceptive and proprioceptive impairments in this population and more specifically underlies dysfunctional multisensory integration. This result could be a first step in the explanation of how visual body image distortions can occur even without a visual deficit and could also explain deficits in implicit body image and body schema found in relation to parietal lobe functioning e.g., [55,57,59,137]. However, as this review found, other research (e.g., [88]) displayed no differences between patients with AN and controls in the ability to examine sensory dimensions relevant to spatial and motion aspects of body-size perception: it is, therefore, necessary to implement further studies in order to shed light on the topic of proprioceptive alterations in EDs and clarify these controversial results.

### 4.3. Vestibular System in Anorexia and Bulimia Nervosa

In this review, the vestibular system appears to be the least investigated dimension of all inner body perceptions in AN and BN. However, vestibular signs play a crucial role in the connection between the spatial description of the inner body and the spatial description of the outside world that allows the development of the allocentric representation of the body [129]. The findings of our review evidence that vestibular signals are impaired in BN but not in patients with AN, suggesting that future studies are needed to deepen the understanding of vestibular signals in EDs. A possible reason behind this is that the reduced postural control could be mostly influenced by musculoskeletal variables. The rapid and significant changes in body weight that are typical of BN may change the proportion of lean-to-fat mass and consequently have an impact on the musculoskeletal system. According to this, body weight variations rather than BMI measurements alone may be responsible for the observed alterations in postural control.

## 5. Conclusions and Future Direction

Deficits in interoception, proprioception, and vestibular signals were observed across AN and BN, suggesting that: (a) alteration of inner body perception might be a crucial feature of EDs, even if further research is needed and (b) VR, to be effective with these patients, has to simulate/modify both the external and the internal body.

First, these results are in agreement with the vision of Riva and colleagues [20,131] who have linked EDs to increased precision of prior body beliefs and/or decreased precision of sensory data, both internal (interoception) and external (proprioception). Moreover, our review reveals a distinction that needs to be further investigated in AN regarding the reliance on vision and proprioception information related to body location and weight estimation. Additional research is needed to know what underlies changes in the processing of visual and proprioceptive signals in EDs and how these modifications may affect the perception of all aspects that rely on visual and proprioceptive signals, such as the location, shape, size, and weight of the body as well as external objects [79,138]. Moreover, the majority of the reviewed articles focused on the AN population instead of BN: this suggests a lack of studies on proprioception and interoception in this clinical population. More research is, therefore, necessary to better understand this aspect and the other dimensions of interoception (cardiac, gastric, and pain) in BN. In addition, there are few studies investigating vestibular signs in EDs. Due to the importance of this system for body perception and body representation in understanding these clinical conditions, further and more sophisticated studies are necessary. Lastly, all the studies we included in our review focus on female patients. Since EDs are a growing phenomenon also among men [139] and gender differences have been found in interoceptive accuracy [140], future studies should focus on male patients with EDs, in order to clarify the role of interoception, proprioception, and vestibular signals in this population, comparing them not only to HCs but also to female patients with EDs to seek for possible differences.

Second, up to now, existing body swapping illusions simulate the external body only, embodying the user in the avatar generated by VR [141]. However, as demonstrated by previous research [33], this approach is not able to permanently correct an impaired body perception in EDs. Why? Our review provides a possible explanation for this: VR simulates and corrects only the representation of the external body (body image) and not the internal one (inner body/body schema), which also apparently plays a critical role in the etiology of EDs. In fact, our experience of the body is the result of the integration of many bodily signals that have to be controlled and matched [34,142]: from (a) outside (exteroception, the body perceived through the senses), from (b) within (inner body), including interoception, the sense of the physiological conditions of the body, proprioception, the sense of the position of the body/body segments and vestibular input, the sense of motion of the body) and from (c) memory [34]. To overcome this problem, we recently suggested a new clinical approach [143]—Regenerative Virtual Medicine (RVM)—that integrates VR with different technology-based somatic modification techniques which are also able to address and modify our inner body experience.

The core elements of RVM are rooted in the Bayesian model of the mind [144,145], which considers the brain as a predictive system that constantly generates probabilistic permutations of its own states in an attempt to maintain a corrected homeostatic balance. From this point of view, an aberration in these predictions, in the past stored models, or in the sensory afferent input, can lead to pathological states and, ultimately, reflect on the body itself. The proposal of RVM suggests that the aberrant contents of pathological bodily representation can be accessed, rewritten, and ultimately modified through the means of technology able to modulate and alter all the components of our body experience. Specifically, this framework suggests using at the same time different technologies—VR, interoceptive technologies, and brain stimulation technologies (see Figure 2)—targeting a different component of our bodily experience to deliver new unexpected healthy probabilistic multisensory representations. A critical role is played by interoceptive technologies [146] for their ability to modulate the inner body. Interoceptive technologies consist of tools that produce direct modulations of interoceptive signals (such as c-fiber stimulation [147,148]), or sonoception [149,150], as well as tools that create illusions by giving people false feedback about their physiological states [151]. Specifically, RVT is based on the following steps: (a) the creation of a synthetic full-body illusion in VR (external body) that is synchronized with an interoceptive modulation (inner body), which can generate considerable prediction error; (b) the use of brain stimulation techniques to lessen the impact of predictions made from the top-down; (c) the application of conscious awareness to increase the accuracy of the multisensory experience; (d) reconstructing and re-explaining the emotional content of the multisensory experience to increase its level of reward using cognitive reappraisal. This process should lead the brain to activate internal regenerative processes able to rewrite the pathological condition and trigger a healing response [143]. Unfortunately, at the moment RVT is just a new method based on the principles of computational neuroscience and not a validated approach. Future studies and clinical trials are required for considering RVT as a possible alternative to the methods used by psychiatry and psychotherapy in the treatment of EDs. To support the use of RVT, artificial intelligence [142] could be particularly useful to integrate information coming from social network systems (SNS). SNS expose people to social comparison; when people with EDs contact with their ideal bodies or shapes in SNS (e.g., social media), they are more likely to feel negative emotions that hinder their motivations to make a change toward healthier bodies. In this sense, natural language processing [152,153] could be used to monitor patients undergoing RVT to identify those at risk of suicide [154], in need of psychological support [155], as well as to oversee the evolution of symptoms and the severity of the pathology.

## Figures and Tables

**Figure 1 jcm-11-07134-f001:**
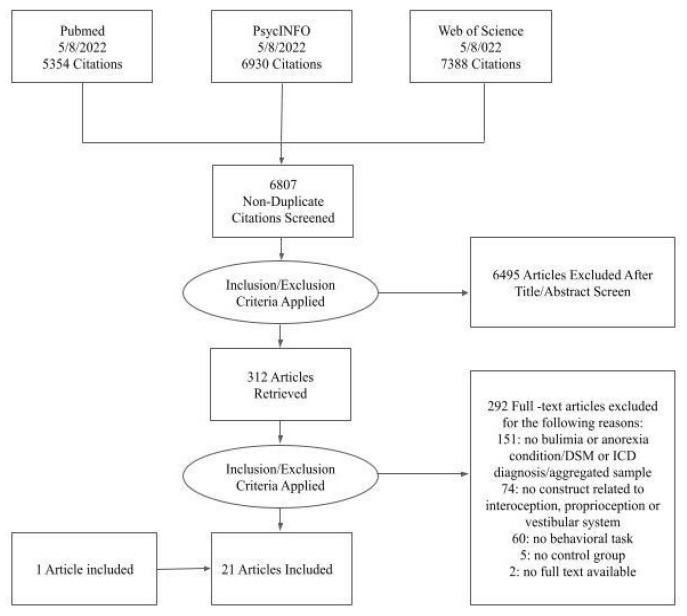
Flow chart of the systematic review. The figure illustrates the search strategy of the systematic review conducted under PRISMA guidelines.

**Figure 2 jcm-11-07134-f002:**
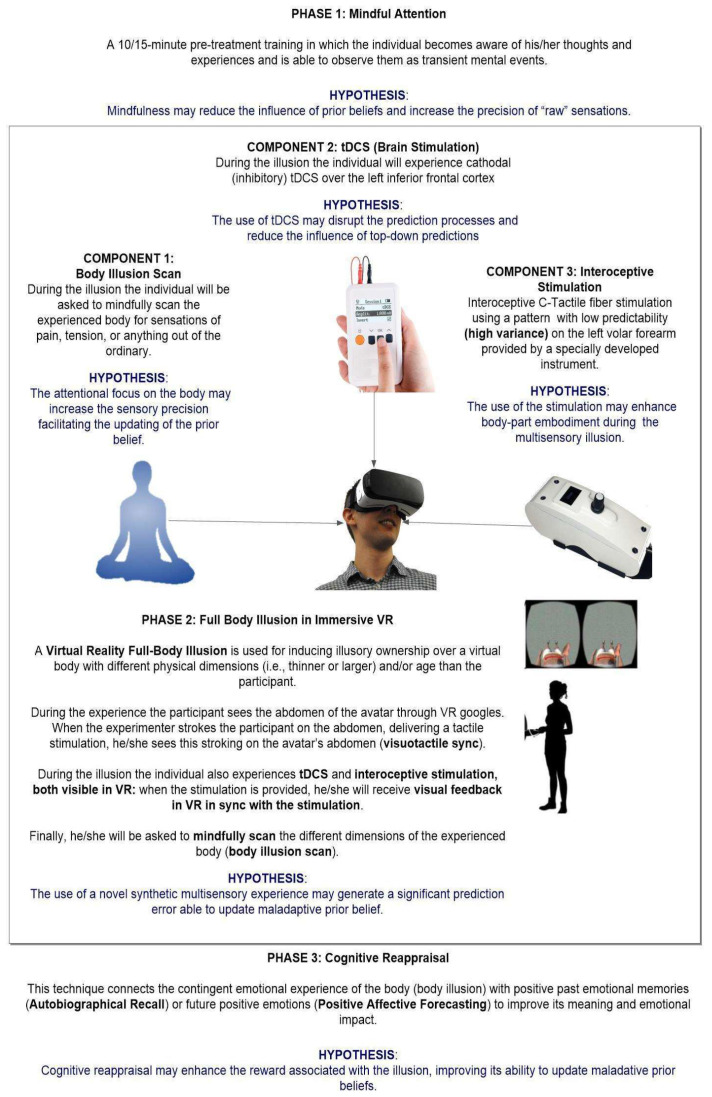
The different tools used in Regenerative Virtual Therapy (Adapted from Riva et al., 2021 [143]). tDCS: transcranial Direct Current Stimulation; VR: Virtual Reality.

**Table 1 jcm-11-07134-t001:** Studies characteristics according to extraction parameters.

Authors and Year	Sample	Gender	Diagnosis	Task/Instrument	Constructs Measured	Primary Outcomes
(Ambrosecchia et al., 2017 [74])	AN: 24 HC: 25	F	AN	Heartbeat perception task	Interoception	Results showed no differences between AN and HCs in heartbeat perception task.
(Aschenbrenner et al., 2009 [75])	AN:16 BN: 24 HC: 23	F	AN BN	“Sniffin’ Sticks” Test Battery and ‘‘Taste Strip” Test Kit	Interoception	Compared to HC and BN, individuals with AN showed lowered olfactory and gustatory sensitivities.
(Bär et al., 2006 [76])	AN: 15 HC: 15	F	AN	Heat Pain Thresholds	Interoception	The heat pain thresholds were significantly increased in the acute state of AN and decreased after weight had been regained for 6 months.
(Bellard et al., 2022 [77])	AN: 27 RAN: 29 HC: 35	F	AN	Affective touch	Interoception	AN and RAN did not differ in their pleasantness ratings to touch for another compared to HC, but when evaluating touch for self, both AN and RAN rated CT-optimal touch as less pleasant than HCs.
(Brown et al., 2022 [78])	AN: 10 HC: 10	F	AN	Behavioral Water Load Task	Interoception	Participants with AN drank significantly less water than HC, but reported greater increases in negative affects pre-to-post-Water Load Task.
(Case et al., 2011 [79])	AN:10 HC: 10	F	AN	Size Weight Illusion	Proprioception	Results showed a reduction in size weight illusion in individuals with AN compared to controls.
(Crucianelli et al. 2016 [80])	AN: 25 HC: 30	F	AN	Affective touch	Interoception	Results showed less pleasure in people with AN regarding affective touch compared to HCs.
(Crucianelli et al., 2020 [81])	AN: 27 RAN: 24 HC: 27	F	AN	Affective Touch	Interoception	Both AN and RAN anticipated tactile experiences and rated delivered tactile stimuli as less pleasant than HCs.
(Demartini et al., 2017 [82])	AN: 20 FMS: 20 HC: 20	F	AN; FMCS (Functional Motor Symptoms)	Heartbeat Perception Task	Interoception	Results showed no differences between people with AN and HC in interoceptive sensitivity and interoceptive awareness.
(Di Lernia et al., 2019 [83])	AN: 1 (single case) HC: 4	F	AN	Heartbeat Perception Task; Metacognitive Confidence in Heartbeat Task Perception; Interoceptive Buffer Saturation Index	Interoception	The patient with AN showed a dissociation of interceptive axes with widespread perceptional deficits.
(Epstein et al., 2001 [84])	AN:20 HC: 20	F	AN	‘‘Proprioception Test’’ and ‘‘Right-Left Orientation Test’’	Proprioception	People with AN showed significantly lower scores in the ‘‘right-left orientation test’’ at pre-treatment assessment as compared to HCs.
(Fontana et al., 2009 [85])	AN: 15 BN: 15 HC: 11	F	AN BN	Kinematics (or segmental) Method	Vestibular Signals	Patients with BN were more unstable than HCs, showing significant differences in anteroposterior center of mass (CoM) excursions and length of the path, while individuals with AN showed no significant differences from HCs.
(Goldzak-Kunik et al., 2012 [71])	AN: 15 HC: 15	F	AN	Interoception: Cold Pain, VAS for Cold, Unpleasantness, and Pain. Proprioception: Kinesthesia task	Interoception and Proprioception	Patients with AN and HCs did not differ in cold pain responses and at the kinaesthesia task.
(Kinnaird et al., 2020 [86])	AN: 37 HC: 37	F	AN	Heartbeat Perception Task; Metacognitive Confidence in Heartbeat Task Perception	Interoception	Heartbeat perception performance was not found to be altered in the AN group compared to the HC group. However, confidence ratings in task performance in the AN group were lower compared to the HC group.
(Lutz et al., 2019 [87])	AN: 20 HC: 20	F	AN AN	Heartbeat Perception Task; Interoceptive Sensibility Task	Interoception	Results showed that people with AN and HCs did not differ significantly in interoceptive accuracy or interoceptive sensibility.
(Mergen et al., 2018 [88])	AN: 27 HC: 40	F	AN	One-Point-Localization Task	Proprioception	Results showed no difference between AN and HC in their performance since both groups showed alterations in the localization task.
(Pollatos et al., 2016 [89])	AN: 15 HC: 15	F	AN	Heartbeat Perception Task	Interoception	During the self-focus, individuals with AN showed lower Interoception accuracy compared to HCs.
(Richard et al., 2019 [90])	AN: 37 HC: 39	F	AN	Heartbeat Perception Task	Interoception	Results showed no evidence of lower heartbeat perception in people with AN compared to HCs.
(Wollast et al., 2022 [91])	AN: 25 HC: 25	F	AN	Heartbeat Perception Task	Interoception	A deficit in interoceptive accuracy was observed for the individuals suffering from AN at rest as well as when an emotional context was induced, compared to HCs.
(Yamamotova et al., 2009 [92])	BN:21 HC: 21	F	BN	Heat Pain Threshold using Analgesia Meterradiant Heat applied to 1 cm^2^	Interoception	BN had a higher pain threshold than HCs in all six conditions. BN also had shorter tolerance latency of cold pressor than HCs.
(Zopf et al., 2016 [93])	AN: 23 HC: 23	F	AN	Rubber Hand Illusion; Proprioception Drift	Proprioception	Results showed the reduced influence of proprioceptive signals on hand location estimates in AN compared to HCs.

AN = Anorexia Nervosa; BN = Bulimia Nervosa; HCs = Healthy Controls; RAN = Recovered from AN; FMS = Functional Motor Symptoms.

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
