# Peer review of "From Virtual Reality to Regenerative Virtual Therapy: Some Insights from a Systematic Review Exploring Inner Body Perception in Anorexia and Bulimia Nervosa"

_jcm, 2022, doi:10.3390/jcm11237134_

Round 1
Reviewer 1 Report
The article deals with an extremely relevant theme in the field of eating disorders, has a well outlined methodology and a detailed discussion of the results, contributing to a good visualization of the current state of research in the area, and also suggesting an innovation for the treatment related to inner body perception in anorexia and bulimia nervosa. I only consider important that, besides the innovative treatment proposal pointed out, other available alternatives (even if partial or still limited) that could already be used to deal with the needs exposed were also pointed out.
Author Response
Thank you very much for your valuable feedback. We described in the manuscript on pages 4 and 5 the important studies that have been carried out using VR (i.e., body illusions, body swapping). VR targets, indeed, the altered body perception of individuals with EDs and has already proven its effectiveness.
Thanks to your comment, however, we had the opportunity to present an additional treatment that is used to intervene in the altered inner body perception of patients with EDs (see pages 7 and 8). Despite several authors suggested the use of neurofeedback (Bartholdy et al., 2013), invasive (Duriez et al., 2020), and non-invasive brain stimulation (Dunlop, Woodside, & Downar, 2016; Duriez et al., 2020) to target the altered inner body perception of individuals with EDs, no trials employing such methodologies have not been implemented on this clinical population, yet. The only technique that has been tested on patients with EDs is cold-water caloric vestibular stimulation (CVS) (Schonherr, & Albrecht May, 2015), widely used in clinical settings to detect balance issues or establish the lack of brainstem function. CVS is a non-invasive brain stimulation technique able to activate key nodes of the anterior cinguloinsular network (aCIN), altered in a wide variety of psychiatric and neurological conditions (e.g., EDs) (Miller, 2016). This technique promotes vestibular neuromodulation by performing a cold-water caloric vestibular stimulation of the ears. Specifically, external auditory canals are warmed or cooled using air or water irrigators. Temperature changes that are both warming and cooling cause the endolymphatic fluid in the semicircular canals to change in density, which in turn causes convection currents that cause cupular deflection, alter the tonic firing rate of the vestibular nerves and cause the vestibulo-ocular reflex or horizontal nystagmus (Been et al., 2007). According to Schonherr and colleagues (2015), after the CVS (on the left and right ears) patients showed significantly smaller estimation of thigh width than before and closer to the real measurement, and the Body-Perception-Index (BPI) reduced dramatically as well.
Reference:
-Bartholdy, S., Musiat, P., Campbell, I. C., & Schmidt, U. (2013). The potential of neurofeedback in the treatment of eating disorders: A review of the literature. European Eating Disorders Review, 21(6), 456-463.
-Duriez, P., Bou Khalil, R., Chamoun, Y., Maatoug, R., Strumila, R., Seneque, M., ... & Guillaume, S. (2020). Brain stimulation in eating disorders: state of the art and future perspectives. Journal of clinical medicine, 9(8), 2358.
-Dunlop, K. A., Woodside, B., & Downar, J. (2016). Targeting neural endophenotypes of eating disorders with non-invasive brain stimulation. Frontiers in Neuroscience, 10, 30.
-tms etc efficaci nello stimolare aree identificate come importanti per le alterazioni dei disturbi relativi alla propriocezione in ED.
-Miller, S. M. (2016). Vestibular neuromodulation: stimulating the neural crossroads of psychiatric illness. Bipolar Disorders, 18(6), 539-543.
-Schonherr, A. and Albrecht May, C. (2015). Influence of Caloric Vestibular Stimulation on Body Experience of Anorectic Patients. Case Studies Journal. ISSN (2305-509X) –2015 Volume 4, Issue 1, Available at SSRN: https://ssrn.com/abstract=2886236.
-Been G, Ngo TT, Miller SM, Fitzgerald PB. (2007). The use of tDCS and CVS as methods of non-invasive brain stimulation. Brain Res Rev. 56(2):346-61. doi: 10.1016/j.brainresrev.2007.08.001. Epub 2007 Aug 28. PMID: 17900703.
Reviewer 2 Report
Thanks for authors to provide a chance to review the paper. It's a very new and interesting topic in the filed of AN treatment, and provides a full picture about the VR and VT in the intervention program. I also have some suggestions to improve the paper as the follows.
1、It's better to add the quantity analysis based the reliable or recruited articles as mentioned in the method part.
2、We need more information about the VR and VT, especially the difference or superiority of VT in treatment of AN, BE,ED. Moreover, the advantage of VR or VT, as compared to traditional treatment should be more clear in the review part.
3、As we known, the digital screen, such as SNS is an important platform of social compare with others. Possibly, when people with AN or ED are more frequent in contacting with ideal body or shapes, they are likely to feel more negative emotions that hinder their motivations to make a change in body shapes.
Author Response
Revision n.1. It's better to add the quantity analysis based on the reliable or recruited articles as mentioned in the method part.
Answer: Thank you very much for your comment. The guidelines we used for this systematic review are the Preferred Reporting Items for Systematic Reviews and Meta-Analysis (PRISMA) guidelines (Moher et al., 2009), which are employed both for systematic reviews and meta-analysis. In the current work, we decided to focus on the systematic review only: a recent meta-analysis was indeed recently realized by Jenkinson et al. (2018), and for avoiding redundancies between our studies, we opted not to include the meta-analysis in our work. However, thanks to your valuable feedback, a meta-analysis on the topic could be an interesting future direction to pursue.
Reference: -Jenkinson, P. M., Taylor, L., & Laws, K. R. (2018). Self-reported interoceptive deficits in eating disorders: A meta-analysis of studies using the eating disorder inventory. Journal of psychosomatic research, 110, 38-45.
Revision n. 2 We need more information about VR and VT, especially the difference or superiority of VT in the treatment of AN, BE, and ED. Moreover, the advantage of VR or VT, as compared to traditional treatment should be more clear in the review part.
Answer: Thank you very much for your comment. VR has been extensively used in treating EDs for its ability to simulate real-world situations, under-regulated and ecological circumstances (Ferrer-García, & Gutiérrez-Maldonado, 2012). In comparison to in vivo exposure, VR offers a higher level of control and safety, permits the inclusion of contextual and proximal cues, prevents unforeseen events during exposure, and helps to customize exposure to the needs of each patient, thereby lowering any treatment resistance and boosting motivation. In vivo exposure can indeed be occasionally complex, for example, due to the challenges with upholding the required standards of safety and confidentiality when exposure is undertaken in a real-world setting, travel time to the exposure location such as the person’s kitchen, or poor control over the stimuli (Ferrer-García, & Gutiérrez-Maldonado, 2012). These restrictions can be somewhat circumvented by doing in vivo exposure in the clinic, although this method only permits exposure to proximate signals (such as meals), not to contextual cues (e.g., kitchen). Imagery exposure is another option. However, if on the one hand, it addresses some of the aforementioned drawbacks, on the other hand, it also requires a significant amount of cognitive effort and may exhaust patients. As a result, there is a higher chance that patients will use avoidance tactics because clinicians cannot fully control the scenario that patients are imagining (Ferrer-García, & Gutiérrez-Maldonado, 2012). When compared to imagery exposure, VR stimulates a variety of sensory modalities (e.g., auditory and visual), making it easier for participants who have trouble picturing scenes to participate. Additionally, since therapists can see what the patient is seeing at any given time, VR aids in the identification of the stimuli that are triggering a given emotional response (Ferrer-García, & Gutiérrez-Maldonado, 2012). VR-based cue exposure therapy (VR-CET) has proven greater effectiveness than cognitive behavior therapy (CBT) as a second-level treatment to decrease binge and purge episodes in individuals with bulimia nervosa (BN) and binge-eating disorder (BED), showing a higher reduction in overeating episodes, and a decrease in binge abstinence rates as well (Ferrer-Garcia et al., 2019). These findings are also confirmed by another study that evaluated the efficacy of VR-CET as a second-level treatment for BN and BED patients (Ferrer‐García et al., 2017). The results of this work revealed that a better overall short-term outcome (i.e., at post-treatment) was observed in the VR-CET group, with a significantly higher reduction in the number of binge and purge episodes, binge-purging abstinence rates and self-reported tendency to engage in overeating episodes, food craving, and anxiety than the CBT group. Similar outcomes have been reported by Marco, Perpiñá, and Botella (2013). In their study, patients who received VR+CBT improved more than the group who did not receive the VR component (i.e., CBT only), also at one-year follow-up (Marco, Perpiñá, & Botella, 2013).
We added this section to pages 3 and 4.
References: -Ferrer-García, M., & Gutiérrez-Maldonado, J. (2012). The use of virtual reality in the study, assessment, and treatment of body image in eating disorders and nonclinical samples: a review of the literature. Body image, 9(1), 1-11.
-Ferrer-Garcia, M., Pla-Sanjuanelo, J., Dakanalis, A., Vilalta-Abella, F., Riva, G., Fernandez-Aranda, F., ... & Gutiérrez-Maldonado, J. (2019). A randomized trial of virtual reality-based cue exposure second-level therapy and cognitive behavior second-level therapy for bulimia nervosa and binge-eating disorder: Outcome at six-month followup. Cyberpsychology, Behavior, and Social Networking, 22(1), 60-68.
-Ferrer‐García, M., Gutiérrez‐Maldonado, J., Pla‐Sanjuanelo, J., Vilalta‐Abella, F., Riva, G., Clerici, M., ... & Dakanalis, A. (2017). A randomised controlled comparison of second‐level treatment approaches for treatment‐resistant adults with bulimia nervosa and binge eating disorder: Assessing the benefits of virtual reality cue exposure therapy. European Eating Disorders Review, 25(6), 479-490.
-Marco, J. H., Perpiñá, C., & Botella, C. (2013). Effectiveness of cognitive behavioral therapy supported by virtual reality in the treatment of body image in eating disorders: one year follow-up. Psychiatry research, 209(3), 619-625.
Revision n. 3 As we know, the digital screen, such as SNS is an important platform of social comparison with others. Possibly, when people with AN or ED are more frequent in contact with ideal bodies or shapes, they are likely to feel more negative emotions that hinder their motivations to make a change in body shapes.
Answer: Thank you for your comment: we added this part on the page 37, including the references you recommended. Social Network Systems are an important platform for social comparison with others, and when people with AN or EDs contact with their ideal bodies or shapes in SNS (e.g., social media), they are more likely to feel negative emotions that hinder their motivations to make a change towards healthier bodies. Natural Language Processing (NLP) could be a particularly useful application in this context, in that patients’ comments on photos representing their ideal bodies or thinness-oriented contents (e.g., models on fashion pages) could be used to monitor patients’ progress (if the patient is undergoing treatments), to screen possible relapses (if the patient is recovered and out of the hospital system), as well as to identify particularly distressing periods and an aggravation of the pathology (if the patient is not recovered and is still receiving treatments). NLP has been already used in EDs (Cliffe et al., 2021; Barańska et al., 2022; Rojewska et al., 2022): for example, to analyze pro-anorexic and pro-bulimic forums, showing its efficacy in identifying users in need of immediate help from mental health professionals (Yan et al., 2019), as well as to screening people at risk for suicide (Coppersmith et al., 2018). Mortality rates are particularly high among people suffering from an ED (Arcelus, Mitchell, Wales, & Nielsen, 2011), with suicide being one of the main causes of death among these patients (Smith, Zuromski, & Dodd, 2018). For this reason, using NLP to screen individuals with EDs most at risk of suicide (i.e., Coppersmithet al., 2018), those who urgently need psychological support (i.e., Yan et al., 2019) as well as to monitor the evolution of symptoms and the severity of the pathology could be essential from a preventive point of view and as an all-encompassing care of the patient.
References: -Yan, H., Fitzsimmons‐Craft, E. E., Goodman, M., Krauss, M., Das, S., & Cavazos‐Rehg, P. (2019). Automatic detection of eating disorder‐related social media posts that could benefit from a mental health intervention. International Journal of Eating Disorders, 52(10), 1150-1156.
-Coppersmith, G., Leary, R., Crutchley, P., & Fine, A. (2018). Natural language processing of social media as screening for suicide risk. Biomedical informatics insights, 10, 1178222618792860.
-Arcelus, J., Mitchell, A. J., Wales, J., & Nielsen, S. (2011). Mortality rates in patients with anorexia nervosa and other eating disorders: a meta-analysis of 36 studies. Archives of general psychiatry, 68(7), 724-731.
-Smith, A. R., Zuromski, K. L., & Dodd, D. R. (2018). Eating disorders and suicidality: what we know, what we don’t know, and suggestions for future research. Current opinion in psychology, 22, 63-67.
-Cliffe C, Seyedsalehi A, Vardavoulia K, Bittar A, Velupillai S, Shetty H, Schmidt U, Dutta R. Using natural language processing to extract self-harm and suicidality data from a clinical sample of patients with eating disorders: a retrospective cohort study. BMJ Open. 2021 Dec 31;11(12):e053808. doi: 10.1136/bmjopen-2021-053808. PMID: 34972768; PMCID: PMC8720985.
-Barańska, K.; Różańska, A.; Maćkowska, S.; Rojewska, K.; Spinczyk, D. Determining the Intensity of Basic Emotions among People Suffering from Anorexia Nervosa Based on Free Statements about Their Body. Electronics 2022, 11, 138. https://doi.org/10.3390/electronics11010138
-Rojewska, K.; Maćkowska, S.; Maćkowski, M.; Różańska, A.; Barańska, K.; Dzieciątko, M.; Spinczyk, D. Natural Language Processing and Machine Learning Supporting the Work of a Psychologist and Its Evaluation on the Example of Support for Psychological Diagnosis of Anorexia. Appl. Sci. 2022, 12, 4702. https://doi.org/10.3390/app12094702
Reviewer 3 Report
The authors have rightly pointed out the advantages of using VR to support the process of achieving the balance of functioning and the image of the body of people suffering from AN and BN. Direct assessment of the patient's condition is difficult. Therefore, the use of new VR techniques can also be supported by Natural Language Processing (NLP) methods that allow for a quantitative description of the patient's condition using for example free notes written by the patients. In my opinion, it would be worth extending the paper with this information, as an example of non-invasive methods of assessing emotions and body image with the use of NLP. For example, such topics are discussed in the works:
Cliffe C, Seyedsalehi A, Vardavoulia K, Bittar A, Velupillai S, Shetty H, Schmidt U, Dutta R. Using natural language processing to extract self-harm and suicidality data from a clinical sample of patients with eating disorders: a retrospective cohort study. BMJ Open. 2021 Dec 31;11(12):e053808. doi: 10.1136/bmjopen-2021-053808. PMID: 34972768; PMCID: PMC8720985.
Barańska, K.; Różańska, A.; Maćkowska, S.; Rojewska, K.; Spinczyk, D. Determining the Intensity of Basic Emotions among People Suffering from Anorexia Nervosa Based on Free Statements about Their Body. Electronics 2022, 11, 138. https://doi.org/10.3390/electronics11010138
Rojewska, K.; Maćkowska, S.; Maćkowski, M.; Różańska, A.; Barańska, K.; Dzieciątko, M.; Spinczyk, D. Natural Language Processing and Machine Learning Supporting the Work of a Psychologist and Its Evaluation on the Example of Support for Psychological Diagnosis of Anorexia. Appl. Sci. 2022, 12, 4702. https://doi.org/10.3390/app12094702
Author Response
Answer: Thank you for your comment: we added this part on page 37, including the references you recommended. Social Network Systems are an important platform for social comparison with others, and when people with AN or EDs contact with their ideal bodies or shapes in SNS (e.g., social media), they are more likely to feel negative emotions that hinder their motivations to make a change towards healthier bodies. Natural Language Processing (NLP) could be a particularly useful application in this context, in that patients’ comments on photos representing their ideal bodies or thinness-oriented contents (e.g., models on fashion pages) could be used to monitor patients’ progress (if the patient is undergoing treatments), to screen possible relapses (if the patient is recovered and out of the hospital system), as well as to identify particularly distressing periods and an aggravation of the pathology (if the patient is not recovered and is still receiving treatments). NLP has been already used in EDs (Cliffe et al., 2021; Barańska et al., 2022; Rojewska et al., 2022): for example, to analyze pro-anorexic and pro-bulimic forums, showing its efficacy in identifying users in need of immediate help from mental health professionals (Yan et al., 2019), as well as to screening people at risk for suicide (Coppersmith et al., 2018). Mortality rates are particularly high among people suffering from an ED (Arcelus, Mitchell, Wales, & Nielsen, 2011), with suicide being one of the main causes of death among these patients (Smith, Zuromski, & Dodd, 2018). For this reason, using NLP to screen individuals with EDs most at risk of suicide (i.e., Coppersmithet al., 2018), those who urgently need psychological support (i.e., Yan et al., 2019) as well as to monitor the evolution of symptoms and the severity of the pathology could be essential from a preventive point of view and as an all-encompassing care of the patient.
References: -Yan, H., Fitzsimmons‐Craft, E. E., Goodman, M., Krauss, M., Das, S., & Cavazos‐Rehg, P. (2019). Automatic detection of eating disorder‐related social media posts that could benefit from a mental health intervention. International Journal of Eating Disorders, 52(10), 1150-1156.
-Coppersmith, G., Leary, R., Crutchley, P., & Fine, A. (2018). Natural language processing of social media as screening for suicide risk. Biomedical informatics insights, 10, 1178222618792860.
-Arcelus, J., Mitchell, A. J., Wales, J., & Nielsen, S. (2011). Mortality rates in patients with anorexia nervosa and other eating disorders: a meta-analysis of 36 studies. Archives of general psychiatry, 68(7), 724-731.
-Smith, A. R., Zuromski, K. L., & Dodd, D. R. (2018). Eating disorders and suicidality: what we know, what we don’t know, and suggestions for future research. Current opinion in psychology, 22, 63-67.
-Cliffe C, Seyedsalehi A, Vardavoulia K, Bittar A, Velupillai S, Shetty H, Schmidt U, Dutta R. Using natural language processing to extract self-harm and suicidality data from a clinical sample of patients with eating disorders: a retrospective cohort study. BMJ Open. 2021 Dec 31;11(12):e053808. doi: 10.1136/bmjopen-2021-053808. PMID: 34972768; PMCID: PMC8720985.
-Barańska, K.; Różańska, A.; Maćkowska, S.; Rojewska, K.; Spinczyk, D. Determining the Intensity of Basic Emotions among People Suffering from Anorexia Nervosa Based on Free Statements about Their Body. Electronics 2022, 11, 138. https://doi.org/10.3390/electronics11010138
-Rojewska, K.; Maćkowska, S.; Maćkowski, M.; Różańska, A.; Barańska, K.; Dzieciątko, M.; Spinczyk, D. Natural Language Processing and Machine Learning Supporting the Work of a Psychologist and Its Evaluation on the Example of Support for Psychological Diagnosis of Anorexia. Appl. Sci. 2022, 12, 4702. https://doi.org/10.3390/app12094702